# NOVELTY DETECTION VIA BLURRING

**Sungik Choi & Sae-Young Chung**
School of Electrical Engineering
Korea Advanced Institute of Science and Technology
Daejeon, Republic of Korea
{si_choi,schung}@kaist.ac.kr

## ABSTRACT

Conventional out-of-distribution (OOD) detection schemes based on variational autoencoder or Random Network Distillation (RND) have been observed to assign lower uncertainty to the OOD than the target distribution. In this work, we discover that such conventional novelty detection schemes are also vulnerable to the blurred images. Based on the observation, we construct a novel RND-based OOD detector, SVD-RND, that utilizes blurred images during training. Our detector is simple, efficient at test time, and outperforms baseline OOD detectors in various domains. Further results show that SVD-RND learns better target distribution representation than the baseline RND algorithm. Finally, SVD-RND combined with geometric transform achieves near-perfect detection accuracy on the CelebA dataset.

## 1 INTRODUCTION

Out-of distribution (OOD) or novelty detection aims to distinguish samples in unseen distribution from the training distribution. A majority of novelty detection methods focus on noise filtering or representation learning. For example, we train an autoencoder to learn a mapping from the data to the bottleneck layer and use the bottleneck representation or reconstruction error to detect the OOD (Sakruada et al., 2014; Pidhorskyi et al., 2018). Recently, deep generative models (Kingma et al., 2014; Dinh et al., 2017; Goodfellow et al., 2014; Kingma et al., 2018; Schlegl et al., 2017) are widely used for novelty detection due to their ability to model high dimensional data. However, OOD detection performance of deep generative models has been called into question since they have been observed to assign a higher likelihood to the OOD data than the training data (Nalisnick et al., 2019; Choi et al., 2018).

On the other hand, adversarial examples are widely employed to fool the classifier, and training classifiers against adversarial attacks has shown effectiveness in detecting unknown adversarial attacks (Tramer et al., 2018). In this work, we propose blurred data as adversarial examples. When we test novelty detection schemes on the blurred data generated by Singular Value Decomposition (SVD), we found that the novelty detection schemes assign higher confidence to the blurred data than the original data.

Motivated by this observation, we employ blurring to prevent the OOD detector from overfitting to low resolution. We propose a new OOD detection model, SVD-RND, which is trained using the idea of Random Network Distillation (RND) (Burda et al., 2019) to discriminate the training data from their blurred versions. SVD-RND is evaluated in some challenging scenarios where vanilla generative models show nearly 50% detection accuracy, such as CIFAR-10 vs. SVHN and ImageNet vs. CIFAR-10 (Nalisnick et al., 2019). Compared to conventional baselines, SVD-RND shows a significant performance gain from 50% to over 90% in these domains. Moreover, SVD-RND shows improvements over baselines on domains where conventional OOD detection schemes show moderate results, such as CIFAR-10 vs. LSUN.

## 2 PROBLEM FORMULATION AND RELATED WORK

The goal of OOD detection is to determine whether the data is sampled from the target distribution $D$. Therefore, based on the training data $D_{\mathrm{train}} \subset D$, we train a scalar function that expresses the

confidence, or uncertainty of the data. The performance of the OOD detector is tested on the $D_{\text{test}} \subset D$ against the OOD dataset $D_{\text{OOD}}$. We denote an in-distribution data and OOD pair as In : Out in this paper, e.g., CIFAR-10 : SVHN.

In this section, we mention only closely related work to our research. For a broader survey on deep OOD detection, we recommend the paper from Chalapathy et al. (2019).

**OOD Detection:** Majority of OOD detection methods rely on a reconstruction error and representation learning. (Ruff et al., 2018) trained a deep neural network to map data into a minimum volume hypersphere. Generative probabilistic novelty detection (GPND) (Pidhorskyi et al., 2018) employed the distance to the latent data manifold as the confidence measure and trained the adversarial autoencoder (AAE) to model the manifold. Deep generative models are widely employed for latent space modeling in OOD detection (Zenati et al., 2018; Sabokrou et al., 2018). However, a recent paper by Nalisnick et al. (2019) discovered that popular deep generative models, such as variational autoencoder (VAE) (Kingma et al., 2014) or GLOW (Kingma et al., 2018), fail to detect simple OOD. While adversarially trained generative models, such as generative adversarial networks (GAN) (Goodfellow et al., 2014) or AAE, are not discussed in Nalisnick et al. (2019), our experiments with GPND show that such models can also struggle to detect such simple OODs.

**OOD Detection with Additional Data:** Some methods try to solve OOD detection by appending additional data or labels for training. Hendrycks et al. (2019) trained the detector with additional outlier data independent of the test OOD data. Ruff et al. (2019) employed semi-supervised learning for anomaly detection in the scenario where we have ground truth information on few training data. Golan et al. (2018) designed geometrically transformed data and trained the classifier to distinguish geometric transforms, such as translation, flipping, and rotation. Shalev et al. (2018) fine-tuned the image classifier to predict word embedding. However, the intuition behind these methods is to benefit from potential side information, while our self-generated blurred image focuses on compensating for the deep model's vulnerability to OOD data with a lower effective rank.

**Adversarial Examples and Labeled Data:** Some methods combine OOD detection with classification, resulting in OOD detection utilizing labeled data. Adversarial examples can be viewed as generated OOD data that attacks the confidence of a pretrained classifier. Therefore, they share some similarities. For example, Hendrycks et al. (2017) set the confidence as the maximum value of the probability output, which is vulnerable against the adversarial examples generated by the Fast Sign Gradient Method (FSGM) (Goodfellow et al., 2014). On the other hand, Liang et al. (2018) employed FSGM counterintuitively to shift the OOD data from the target further, therefore improving OOD detection. Lee et al. (2018) employed Mahalanobis distance to measure uncertainty in the hidden features of the network, which also proved efficient in adversarial defense.

**Bayesian Uncertainty Calibration:** Bayesian formulation is widely applied for better calibration of the model uncertainty. Recent works employed Bayesian neural networks (Sun et al., 2017) or interpreted a neural network's architecture in the Bayesian formulation, such as dropout (Gal et al., 2016), and Adam optimizer (Khan et al., 2018). Our baseline, RND (Burda et al., 2019), can be viewed as the Bayesian uncertainty of the model weight under randomly initialized prior (Osband et al., 2018).

## 3 METHODOLOGY

In this section, we motivate our use of blurred data as adversarial examples to conventional deep OOD detection methods. Motivatied by the observation, we present our proposed algorithm, SVD-RND, and provide intuitions why SVD-RND help OOD detection.

### 3.1 GENERATING BLURRED DATA

In this work, blurred images function as adversarial examples. We directly employ the SVD on the data matrix of each image in the training data and force the bottom non-zero singular values to zero to construct a blurred image. Suppose that data image $d \in D$ consists of multiple channels, where the $j$-th channel has $N_j$ nonzero singular values $\sigma_{j1} \geqslant \sigma_{j2} \geqslant \ldots \sigma_{jN_j} > 0$. Then, the $j$-th channel can be represented as the weighted sum of orthonormal vectors.

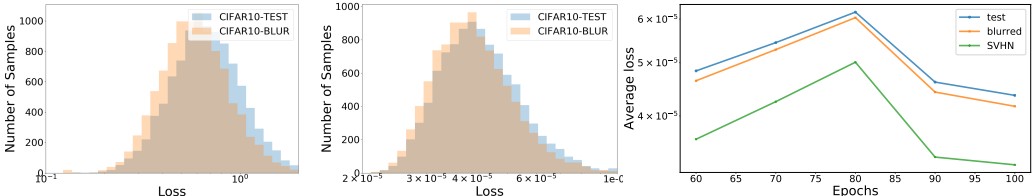

Figure 1: Test loss of VQ-VAE (**left**) and RND (**middle**) on original image and blurred image ($K = 28$) of CIFAR-10 data. RND assigns higher confidence to blurred image and OOD data throughout the training process (**right**).

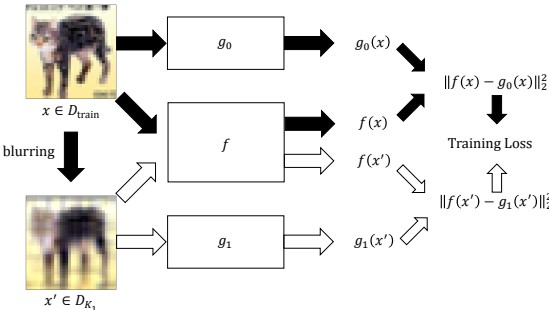

Figure 2: Training of SVD-RND ($b_{\mathrm{train}} = 1$)

$$d_j = \Sigma_{t=1}^{N_j} \sigma_{jt} u_{jt} v_{jt}^T \tag{1}$$

We discard the bottom $K$ non-zero singular values of each channel to construct the blurred image. We test conventional novelty detection methods on blurred images. We first train the VQ-VAE (Oord et al., 2017) in the CIFAR-10 (Krizhevsky et al., 2009) dataset. Figure 1 shows the loss of VQ-VAE on the test data and blurred test data ($K = 28$). We follow the settings of the original paper. VQ-VAE assigns higher likelihood to the blurred data than the original data.

We note that this phenomenon is not constrained to the generative models. We trained the RND on the CIFAR-10 dataset and plot the $L_2$ loss in the test data and blurred test data in Figure 1. We refer to Appendix B for detailed explanation and employed architecture for the RND in the experiment. Furthermore, we plot the average loss on the blurred test data and original test data during the training procedure. Throughout the training phase, the model assigns lower uncertainty to the blurred data. This trend is similar to the CIFAR-10 : SVHN case observed by Nalisnick et al. (2019), where the generative model assigns more confidence to the OOD data throughout the training process.

While we employ SVD for our main blurring technique, conventional techniques in image processing can be applied for blurring, such as Discrete Cosine Transform (DCT) or Gaussian Blurring. However, DCT is quadratic in the size of the hyperparameter search space, therefore much harder to optimize than SVD. We further compare the performance between SVD and other blurring techniques in Section 4.

## 3.2 OOD DETECTION VIA SVD-RND

We now present our proposed algorithm, SVD-RND. SVD-RND trains the predictor network $f$ to discriminate between the original and blurred datasets. We first generate blurred datasets $D_i$ from $D_{\mathrm{train}}$ by zeroing the bottom $K_i$ non-zero singular values of each data channel ($i = 1, \ldots, b_{\mathrm{train}}$, where $b_{\mathrm{train}}$ is the number of generated blurred datasets used for training). We then construct $b_{\mathrm{train}} + 1$ target networks, i.e., $g_0, g_1, \ldots, g_{b_{\mathrm{train}}}$. The target networks are independently randomly initialized and their parameters are unchanged during training. Predictor network $f$ is trained to minimize the $l_2$

Table 1: Experiment In : Out domains

| Target | | OOD | |
|---|---|---|---|
| **CIFAR-10** | SVHN | LSUN | TinyImageNet |
| **TinyImageNet** | SVHN | CIFAR-10 | CIFAR-100 |
| **LSUN** | SVHN | CIFAR-10 | CIFAR-100 |
| **CelebA** | SVHN | CIFAR-10 | CIFAR-100 |

loss against the corresponding target network on each dataset.

$$f^* = \arg\min_f \left[ \Sigma_{x \in D_{\text{train}}} \|f(x) - g_0(x)\|_2^2 + \Sigma_{i=1}^{b_{\text{train}}} \Sigma_{x \in D_i} \|f(x) - g_i(x)\|_2^2 \right] \quad (2)$$

SVD-RND optimizes the predictor network $f$ as shown in (2). When a new test sample $x$ is given, SVD-RND outputs $\|f(x) - g_0(x)\|_2^2$ as the uncertainty of the sample. Figure 2 shows the training process of SVD-RND. While the original RND paper employs a single target network to train the predictor network, SVD-RND employs multiple target networks to discriminate the original data from the blurred images.

While SVD-RND directly regularizes only on the blurred images, OODs can be generated in alternative directions. For completeness, we investigate the performance of conventional models in OODs generated in orthogonal direction to blurring. We refer to Appendix D for the detailed experiment.

## 4 EXPERIMENTAL RESULTS

### 4.1 EXPERIMENT SETTING

SVD-RND is examined in the cases in Table 1. CIFAR-10 : SVHN, CelebA (Liu et al., 2015) : SVHN (Netzer et al., 2011), and TinyImageNet (Deng et al., 2009) : (SVHN, CIFAR-10, CIFAR-100) are the cases studied by Nalisnick et al. (2019). We also studied CIFAR-10 : (LSUN (Yu et al., 2015), TinyImageNet), LSUN : (SVHN, CIFAR-10, CIFAR-100) and CelebA: (CIFAR-10, CIFAR-100) pairs. We implement the baselines and SVD-RND in the PyTorch framework.[1] For a unified treatment, we resized all images in all datasets to $32 \times 32$. We refer to Appendix C for the detailed setting.

For SVD-RND, we optimize the number of discarded singular values over different datasets. We choose the detector with the best performance across the validation data. We refer to Appendix C for the parameter setting. We also examine the case where each image is blurred by DCT and Gaussian blurring. For DCT, we apply the DCT to the image, discard low magnitude signals, and generate the blurred image by inverse DCT. In DCT-RND, we optimize the number of discarded components in the frequency domain. For Gaussian blurring, we optimize the shape of the Gaussian kernel. We denote this method as GB-RND.

We compare the performance of SVD-RND, DCT-RND, and GB-RND to the following baselines.

**Generative Probabilistic Novelty Detector**: GPND (Pidhorskyi et al., 2018) is the conventional generative-model-based novelty detection method that models uncertainty as a deviation from the latent representation, which is modeled by the adversarial autoencoder. We trained GPND with further parameter optimization.

**Geometric Transforms**: Golan et al. (2018) trained a classifier to discriminate in-distribution data against geometrically transformed data to achieve better OOD detection performance. The authors used four types of geometric transforms: flip, rotation, vertical translation, and horizontal translation. We test each transformation by setting them as OOD proxies in the RND framework. Moreover, we also investigate the effect of pixel inversion, contrast reduction, and shearing. We refer to Cubuk et al. (2019) for detailed explanation of the augmentation strategies.

**RND**: We test the original RND (Burda et al., 2019) baseline.

---

[1]Our code is based on `https://github.com/kuangliu/pytorch-cifar`

Table 2: OOD detection results (TNR at 95% TPR) on CIFAR-10, TinyImageNet, LSUN, and CelebA datasets. See Table 1 for OODs used in each case.

| Method | CIFAR-10 | TNR(95% TPR) TinyImageNet | LSUN | CelebA |
|---|---|---|---|---|
| SVD-RND (proposed) | **0.969**/0.956/0.952 | **0.991/0.926/0.911** | **0.995**/0.621/0.614 | **0.999**/0.897/0.897 |
| DCT-RND (proposed) | 0.899/0.797/0.748 | 0.929/0.104/0.169 | 0.971/0.117/0.213 | 0.989/0.491/0.587 |
| GB-RND (proposed) | 0.474/0.803/0.738 | 0.982/0.264/0.321 | 0.986/0.176/0.266 | 0.994/0.455/0.526 |
| RND | 0.008/0.762/0.736 | 0.001/0.001/0.003 | 0.012/0.034/0.075 | 0.067/0.231/0.253 |
| GPND | 0.050/0.767/0.665 | 0.077/0.085/0.118 | 0.051/0.059/0.102 | 0.084/0.230/0.250 |
| Flip | 0.057/0.091/0.081 | 0.160/0.212/0.231 | 0.060/0.055/0.083 | 0.055/0.728/0.750 |
| Rotate | 0.235/0.246/0.308 | 0.711/0.669/0.688 | 0.341/0.278/0.334 | 0.950/**0.937/0.945** |
| Vertical Translation | 0.105/0.649/0.648 | 0.050/0.012/0.012 | 0.117/0.044/0.076 | 0.930/0.887/0.897 |
| Horizontal Translation | 0.070/0.675/0.630 | 0.109/0.005/0.011 | 0.140/0.043/0.101 | 0.894/0.874/0.889 |
| Vertical Shear | 0.077/0.744/0.684 | 0.094/0.058/0.094 | 0.143/0.079/0.134 | 0.940/0.883/0.897 |
| Horizontal Shear | 0.227/0.720/0.672 | 0.072/0.072/0.098 | 0.148/0.084/0.132 | 0.897/0.885/0.888 |
| Contrast | 0.468/0.000/0.002 | 0.670/0.143/0.142 | 0.616/0.123/0.146 | 0.701/0.295/0.329 |
| Invert | 0.473/0.614/0.622 | 0.455/0.057/0.061 | 0.508/0.142/0.172 | 0.766/0.740/0.721 |
| Typicality Test | 0.008/0.734/0.691 | 0.004/0.003/0.008 | 0.004/0.024/0.057 | 0.064//0.245/0.274 |

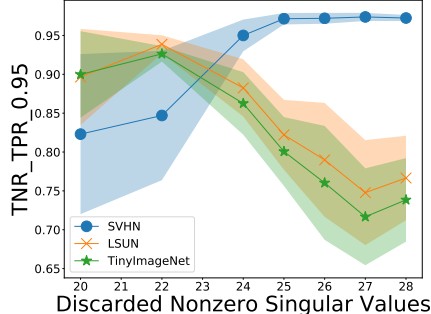 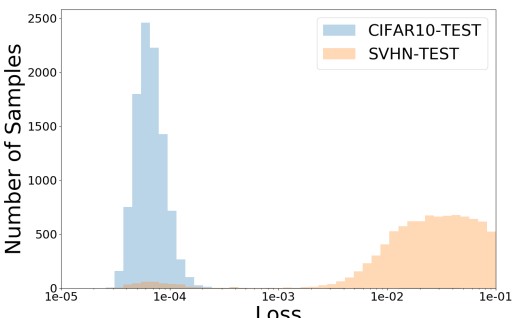

Figure 3: **Left**: Performance of SVD-RND (proposed) for different $K_1$ in CIFAR-10 : (SVHN, LSUN, TinyImageNet) domains. Each filled region is the 95% confidence interval of the detector. SVD-RND shows a small confidence interval in the best performing parameters. **Right**: Histogram of SVD-RND's test loss for CIFAR-10 and SVHN datasets.

**Typicality Test**: Nalisnick et al. (2019) set the OOD metric of the generative model as the distance between the mean log likelihood of the model on the training data and the log likelihood of the model on the test data. We experiment typicality test on the RND framework by employing the test loss of RND instead of log likelihood in the generative models.

Five metrics on binary hypothesis testing are used to evaluate the OOD detectors: area under the Receiver Operating Characteristic curve (AUROC), area of the region under the Precision-Recall curve (AUPR), detection accuracy, and TNR (True negative rate) at 95% TPR (True positive rate). All criteria are bounded between 0 and 1, and the results close to 1 imply better OOD detection.

## 4.2 OOD DETECTION RESULTS

We summarize our results on the TNR at 95% TPR in Table 2. For example, TNR of 96.9% is achieved by SVD-RND for CIFAR-10 : SVHN pair. We refer to Appendix A for the full results. In all In : Out domains except the CelebA : (CIFAR-10, CIFAR-100) domain, SVD-RND outperforms all other baselines in every metric. Furthermore, all the proposed techniques outperform GPND and RND on all In : Out domains. We further visualize the CIFAR-10 data before and after blurring in Appendix E. We plot the performance of SVD-RND over different $K_1$ in Figure 3. In Figure 3, we experimented with 4 seeds. In the best performing parameter for each OOD data, SVD-RND shows stable performance. See Appendix F for results under small $K_1$.

Furthermore, we plot the output of SVD-RND on target CIFAR-10 data and OOD SVHN data when $K_1 = 28$ in Figure 3. Compared to the results in Figure 1, SVD-RND better discriminates SVHN data from the in-distribution data.

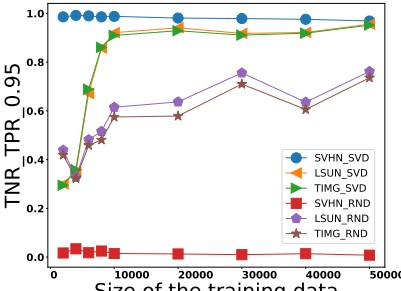 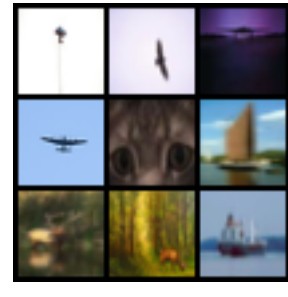 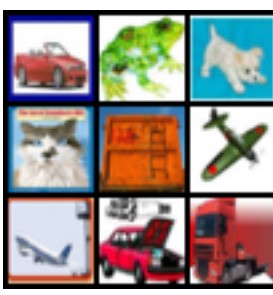

Figure 4: **Left**: Novelty detection performance (TNR at 95% TPR) of SVD-RND and RND on reduced CIFAR-10 training data. SVD-RND is robust to reduced training data while RND's detection performance decreases. **Middle**: Top-9 anomalous CIFAR-10 test samples detected by SVD-RND. **Right**: Top-9 anomalous CIFAR-10 test samples detected by RND.

Table 3: Classification performance of fine-tuned classifier over the activation map trained by SVD-RND, RND, and randomly initialized weights. SVD-RND consistently outperforms RND.

| Activation map | Target: CIFAR-10 | | |
| | SVD-RND ($K$=28) | RND | Random |
| --- | --- | --- | --- |
| 15th (linear) | 55.62 (1.10) | 42.09 (0.66) | 36.55 (0.56) |
| 15th (7-layer) | 86.29 (0.09) | 83.78 (0.29) | 86,56 (0.36) |
| 27th (linear) | 52.69 (0.24) | 38.21 (2.00) | 24.46 (0.42) |
| 27th (7-layer) | 70.31 (0.24) | 57.18 (0.49) | 66.40 (0.31) |

GPND and RND fail to discriminate OOD from the targets in CIFAR-10 : SVHN, LSUN : (SVHN, CIFAR-10, CIFAR-100), TinyImageNet : (SVHN, CIFAR-10, CIFAR-100), and CelebA : SVHN domains. Moreover, GPND performs the SVD of the Jacobian matrix in test time, which makes GPND slower than SVD-RND. Furthermore, we visualize the uncertainty prediction of RND and SVD-RND in Figure 4, which shows the top-9 examples on CIFAR-10 test data, where SVD-RND and RND assign the highest uncertainty. We observe that SVD-RND tends to assign higher uncertainty to blurry or hardly recognizable images compared to RND.

On the other hand, OOD detection schemes based on geometric transformations (Golan et al., 2018) show generally improved results against GPND and RND on detecting OOD data compared to RND and GPND. Especially in CelebA : (SVHN, CIFAR-10, CIFAR-100) domains, rotation-based methods and translation-based methods show excellent performance. However, in the CIFAR-10 target domain, OOD detection schemes based on geometric transformations show degraded performance against RND or GPND on LSUN and TinyImageNet OOD data.

Furthermore, typicality test shows mixed results compared to the baseline algorithms.

Finally, we also investigate the case where limited training data is available. We examined the performance of SVD-RND and RND in CIFAR-10 : (LSUN, TinyImageNet) domains. Figure 4 shows the TNR at 95% TPR metric of each method when the number of training examples is reduced. For each OOD data, we denote the result on SVD-RND as OOD_SVD, and denote the result on RND as OOD_RND.

## 5 FURTHER ANALYSIS

In this section, we examine some other aspects of SVD-RND. In Section 5.1, we examine whether SVD-RND learns better representation compared to the baseline. Furthermore, we propose a novel heuristic for training SVD-RND in Section 5.2, where no validation OOD data is available. Finally, we show that we can further improve the performance of SVD-RND by incorporating geometric transformations in Section 5.3.

Table 4: Performance of uniform SVD-RND and optimized SVD-RND

| Target: CIFAR-10, OOD: SVHN/LSUN/TinyImageNet. Target: TinyImageNet (TIMG), OOD: SVHN/CIFAR-10/CIFAR-100 | | | | | |
|---|---|---|---|---|---|
| Dataset/$b_{\text{train}}$ | AUROC | TNR(95% TPR) | Detection accuracy | AUPR in | AUPR out |
| CIFAR-10/3 (uniform) | 0.967/0.961/0.961 | 0.944/0.827/0.836 | 0.962/0.904/0.904 | 0.843/0.966/0.962 | 0.989/0.949/0.953 |
| CIFAR-10/4 (uniform) | 0.964/0.987/0.988 | 0.941/0.954/0.959 | 0.958/0.957/0.961 | 0.848/0.989/0.989 | 0.987/0.983/0.985 |
| CIFAR-10/1 (optimized) | 0.981/0.985/0.982 | 0.969/0.956/0.952 | 0.980/0.955/0.953 | 0.903/0.987/0.983 | 0.993/0.975/0.976 |
| TIMG/3 (uniform) | 0.993/0.831/0.814 | 0.999/0.745/0.701 | 0.989/0.855/0.832 | 0.991/0.741/0.725 | 0.995/0.878/0.864 |
| TIMG/4 (uniform) | 0.984/0.939/0.923 | 0.954/0.880/0.842 | 0.976/0.927/0.908 | 0.982/0.915/0.894 | 0.989/0.938/0.928 |
| TIMG/2 (optimized) | 0.983/0.969/0.960 | 0.991/0.926/0.911 | 0.980/0.963/0.953 | 0.978/0.965/0.951 | 0.989/0.958/0.953 |

## 5.1 REPRESENTATION LEARNING IN SVD-RND

While SVD-RND outperforms RND on every In : Out domains in Section 4, we provide further evidence that SVD-RND learns superior target distribution representation compared to RND. For the evidence, we fine-tune a classifier over the fixed activation map of SVD-RND and RND. We set the activation map as the output of the first 15 or 27 layers of RND and SVD-RND predictor network trained in CIFAR-10 datasets. For the fine-tuning, we either appended three residual blocks and a linear output layer with softmax activation (denoted as 7-layer in Table 3) or a linear layer (denoted as linear in Table 3). Then, we fine-tune the appended network for the CIFAR-10 classification task. The SGD optimizer with learning rate 0.1 is used for fine-tuning, and the learning rate is annealed from 0.1 to 0.01 and 0.001 after 30 and 60 epochs over 100 epochs of training, respectively. We average the result across three fixed random seeds.

We show our results in Table 3. SVD-RND consistently outperforms RND on the fine-tuning task. Therefore, the result supports that SVD-RND learns better target distribution-specific knowledge.

## 5.2 SVD-RND WITHOUT OOD VALIDATION DATA

In our main experiments in Section 4, we used the OOD validation data for optimizing each $K_i$. However, in realistic scenarios, OOD data are generally unknown to the detector. We propose an effective rank (Roy et al., 2007) based design of SVD-RND that does not use the OOD validation dataset and compare its performance against the results in Section 4. Log effective rank of the single image matrix $D$ is defined as the entropy of the normalized singular values $(\sigma_1, \ldots, \sigma_N)$ of the image matrix.

$$\text{LER}_D = H\left(\frac{\sigma_1}{\Sigma_{j=1}^{N}\sigma_j}, \cdots, \frac{\sigma_N}{\Sigma_{j=1}^{N}\sigma_j}\right) \tag{3}$$

In (3), $H$ is the entropy function. Then, the effective rank is defined as the two to the power of the log effective rank. We set the effective rank of image data as the averaged effective rank of each channel.

Based on (3), we propose selecting each $K_i$ such that average of log effective rank on each blurred dataset is equally spaced. Specifically, suppose the log effective rank of the data averaged in training dataset $D_{\text{train}}$ is $\text{LER}_{D_{\text{train}}}$. Then, we set the target log effective rank $\text{LER}_1, \text{LER}_2, \ldots, \text{LER}_{b_{\text{train}}}$ as follows.

$$\text{LER}_i = \left(0.5 + 0.5 \times \frac{i-1}{b_{\text{train}}}\right)\text{LER}_{D_{\text{train}}} \tag{4}$$

Then, we select $K_i$ such that the average of the log effective rank in the blurred dataset with $K_i$ discarded singular values is closest to $\text{LER}_i$. We test our criterion in CIFAR-10 and TinyImageNet data with different $b_{\text{train}}$. We train SVD-RND for 25 epochs for $b_{\text{train}} = 3$, and 20 epochs for $b_{\text{train}} = 4$. We show the performance of SVD-RND based on (4) in Table 4, which is denoted as SVD-RND (uniform). We also show results of SVD-RND optimized with the validation OOD data from Table 2 and denote them as SVD-RND (optimized) in Table 4. Uniform SVD-RND already outperforms the second-best methods in Table 2. Furthermore, as $b_{\text{train}}$ increases, uniform SVD-RND approaches the performance of the optimized SVD-RND.

Table 5: OOD detection performance of SVD-ROT-RND and SVD-VER-RND

| | | Target: CelebA, OOD: SVHN/CIFAR-10/CIFAR-100 | | | |
|---|---|---|---|---|---|
| Method | AUROC | TNR(95 % TPR) | Detection accuracy | AUPR in | AUPR out |
| SVD-ROT-RND | 0.997/**0.996/0.996** | **0.999/0.993/0.994** | 0.996/**0.991/0.991** | 0.998/**0.998/0.998** | 0.993/**0.986/0.988** |
| SVD-VER-RND | **0.999**/0.993/0.994 | **0.999**/0.982/0.982 | **0.998**/0.982/0.981 | **0.999**/0.997/0.997 | **0.998**/0.984/0.986 |
| SVD-RND | **0.999**/0.963/0.964 | **0.999**/0.897/0.897 | **0.998**/0.928/0.928 | **0.999**/0.981/0.981 | **0.998**/0.941/0.943 |
| ROT-RND | 0.974/0.979/0.982 | 0.950/0.937/0.945 | 0.964/0.952/0.956 | 0.950/0.989/0.991 | 0.981/0.964/0.969 |
| VER-RND | 0.964/0.961/0.964 | 0.930/0.887/0.897 | 0.952/0.923/0.926 | 0.934/0.979/0.980 | 0.975/0.941/0.946 |

## 5.3 FURTHER IMPROVEMENT OF SVD-RND

While SVD-RND achieves reasonable OOD detection performance, combining SVD-RND with other baseline algorithms may further enhance the performance. For example, as shown in Table 2, training against rotated data benefits OOD detection in CelebA dataset. Therefore, we combine SVD-RND and geometric transform-based methods to further improve SVD-RND. We treat both blurred data and geometrically transformed data as OOD and train the predictor network to discriminate the original data from the OOD. We combine rotation and vertical translation with SVD-RND and denote them as SVD-ROT-RND and SVD-VER-RND, respectively.

We compare the performance of SVD-ROT-RND and SVD-VER-RND against rotation and vertical translation in CelebA : (SVHN, CIFAR-10, CIFAR-100) domains. We refer readers to the results in Table 5. We observe that SVD-ROT-RND and SVD-VER-RND outperform their counterparts and SVD-RND. Especially, SVD-ROT-RND and SVD-VER-RND show significant performance gains in CelebA : (CIFAR-10, CIFAR-100) domains.

## 6 CONCLUSION

In this work, blurred images are introduced as adversarial examples in deep OOD detection. SVD-RND is employed for adversarial defense against blurred images. SVD-RND achieves significant performance gain in all In : Out domains. Even without the validation OOD data, we can design SVD-RND to outperform conventional OOD detection models.

## 7 ACKNOWLEDGEMENT

This work was supported by Samsung Electronics and the ICT R&D program of MSIP/IITP. [2016-0-00563, Research on Adaptive Machine Learning Technology Development for Intelligent Autonomous Digital Companion]

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

# A FULL OOD DETECTION RESULTS

Table 6: OOD detection results on CIFAR-10, TinyImageNet, LSUN, and CelebA datasets

| Target: CIFAR-10, OOD: SVHN/LSUN/TinyImageNet | | | | | |
|---|---|---|---|---|---|
| Method | AUROC | TNR(95% TPR) | Detection accuracy | AUPR in | AUPR out |
| SVD-RND (proposed) | **0.981/0.985/0.982** | **0.969/0.956/0.952** | **0.980/0.955/0.953** | **0.903/0.987/0.983** | **0.993/0.975/0.976** |
| DCT-RND (proposed) | 0.944/0.948/0.925 | 0.899/0.797/0.748 | 0.940/0.883/0.861 | 0.769/0.945/0.909 | 0.981/0.946/0.930 |
| GB-RND (proposed) | 0.624/0.952/0.923 | 0.474/0.803/0.739 | 0.722/0.887/0.858 | 0.311/0.950/0.908 | 0.860/0.952/0.928 |
| RND | 0.211/0.941/0.923 | 0.008/0.762/0.736 | 0.500/0.873/0.857 | 0.180/0.937/0.908 | 0.560/0.943/0.931 |
| GPND | 0.230/0.941/0.895 | 0.050/0.767/0.665 | 0.513/0.876/0.828 | 0.190/0.936/0.872 | 0.605/0.941/0.905 |
| Flip | 0.490/0.616/0.607 | 0.057/0.091/0.081 | 0.534/0.601/0.599 | 0.281/0.663/0.656 | 0.707/0.564/0.553 |
| Rotate | 0.853/0.777/0.824 | 0.235/0.246/0.308 | 0.826/0.714/0.755 | 0.735/0.806/0.840 | 0.911/0.719/0.773 |
| Vertical Translation | 0.276/0.924/0.896 | 0.105/0.649/0.648 | 0.540/0.849/0.823 | 0.193/0.923/0.881 | 0.654/0.919/0.899 |
| Horizontal Translation | 0.279/0.917/0.890 | 0.070/0.675/0.630 | 0.523/0.844/0.818 | 0.193/0.915/0.874 | 0.637/0.905/0.889 |
| Vertical Shear | 0.331/0.937/0.909 | 0.077/0.744/0.684 | 0.520/0.869/0.839 | 0.205/0.936/0.896 | 0.663/0.927/0.905 |
| Horizontal Shear | 0.503/0.929/0.902 | 0.227/0.720/0.672 | 0.590/0.860/0.834 | 0.261/0.925/0.886 | 0.781/0.928/0.908 |
| Contrast | 0.659/0.859/0.829 | 0.468/0.000/0.002 | 0.725/0.815/0.785 | 0.331/0.885/0.842 | 0.859/0.746/0.721 |
| Invert | 0.611/0.911/0.900 | 0.473/0.614/0.622 | 0.712/0.849/0.841 | 0.303/0.919/0.897 | 0.848/0.871/0.870 |
| Typicality Test | 0.411/0.929/0.905 | 0.008/0.734/0.691 | 0.501/0.861/0.841 | 0.258/0.917/0.880 | 0.632/0.933/0.916 |

| Target: TinyImageNet, OOD: SVHN/CIFAR-10/CIFAR-100 | | | | | |
|---|---|---|---|---|---|
| Method | AUROC | TNR(95% TPR) | Detection accuracy | AUPR in | AUPR out |
| SVD-RND (proposed) | 0.983/**0.969/0.960** | **0.991/0.926/0.911** | 0.980/**0.963/0.953** | **0.978/0.965/0.951** | 0.989/**0.958/0.953** |
| DCT-RND (proposed) | 0.950/0.317/0.403 | 0.929/0.104/0.169 | 0.958/0.541/0.569 | 0.758/0.404/0.438 | 0.984/0.441/0.518 |
| GB-RND (proposed) | **0.993**/0.497/0.551 | 0.982/0.264/0.321 | **0.991**/0.616/0.643 | 0.969/0.492/0.522 | **0.998**/0.606/0.655 |
| RND | 0.079/0.184/0.213 | 0.001/0.001/0.003 | 0.500/0.500/0.500 | 0.163/0.363/0.371 | 0.513/0.316/0.324 |
| GPND | 0.256/0.367/0.395 | 0.077/0.085/0.118 | 0.514/0.520/0.536 | 0.190/0.424/0.436 | 0.630/0.434/0.473 |
| Flip | 0.550/0.550/0.569 | 0.160/0.212/0.231 | 0.636/0.620/0.625 | 0.294/0.519/0.533 | 0.765/0.569/0.591 |
| Rotate | 0.845/0.806/0.821 | 0.711/0.669/0.688 | 0.868/0.822/0.832 | 0.541/0.727/0.742 | 0.933/0.823/0.841 |
| Vertical Translation | 0.131/0.185/0.213 | 0.050/0.012/0.012 | 0.521/0.502/0.501 | 0.171/0.362/0.370 | 0.567/0.323/0.329 |
| Horizontal Translation | 0.210/0.184/0.224 | 0.109/0.005/0.011 | 0.548/0.500/0.052 | 0.182/0.362/0.375 | 0.627/0.317/0.334 |
| Vertical Shear | 0.330/0.398/0.447 | 0.094/0.058/0.094 | 0.523/0.514/0.525 | 0.204/0.443/0.467 | 0.675/0.437/0.487 |
| Horizontal Shear | 0.296/0.408/0.449 | 0.072/0.072/0.098 | 0.512/0.519/0.527 | 0.197/0.447/0.468 | 0.648/0.453/0.490 |
| Contrast | 0.739/0.337/0.361 | 0.670/0.143/0.142 | 0.824/0.561/0.559 | 0.388/0.412/0.422 | 0.909/0.465/0.475 |
| Invert | 0.561/0.248/0.275 | 0.455/0.057/0.061 | 0.705/0.512/0.514 | 0.285/0.381/0.390 | 0.828/0.365/0.381 |
| Typicality Test | 0.285/0.262/0.285 | 0.004/0.003/0.008 | 0.500/0.500/0.500 | 0.214/0.400/0.409 | 0.577/0.338/0.347 |

| Target: LSUN, OOD: SVHN/CIFAR-10/CIFAR-100 | | | | | |
|---|---|---|---|---|---|
| Method | AUROC | TNR(95% TPR) | Detection accuracy | AUPR in | AUPR out |
| SVD-RND (proposed) | 0.986/**0.795/0.801** | **0.995/0.621/0.614** | 0.983/**0.787/0.783** | **0.975/0.724/0.730** | 0.990/**0.828/0.834** |
| DCT-RND (proposed) | 0.984/0.508/0.575 | 0.971/0.117/0.213 | 0.981/0.535/0.583 | 0.920/0.513/0.552 | 0.995/0.534/0.621 |
| GB-RND (proposed ) | **0.993**/0.538/0.601 | 0.986/0.176/0.266 | **0.989**/0.566/0.609 | 0.967/0.534/0.570 | **0.997**/0.580/0.656 |
| RND | 0.190/0.430/0.467 | 0.012/0.034/0.075 | 0.500/0.500/0.514 | 0.177/0.476/0.489 | 0.557/0.427/0.479 |
| GPND | 0.250/0.459/0.487 | 0.051/0.059/0.102 | 0.513/0.509/0.529 | 0.192/0.486/0.495 | 0.611/0.462/0.509 |
| Flip | 0.438/0.486/0.507 | 0.060/0.055/0.083 | 0.524/0.508/0.522 | 0.249/0.511/0.525 | 0.685/0.468/0.500 |
| Rotate | 0.909/0.752/0.779 | 0.341/0.278/0.334 | 0.889/0.736/0.764 | 0.807/0.700/0.721 | 0.943/0.743/0.778 |
| Vertical Translation | 0.258/0.415/0.446 | 0.117/0.044/0.076 | 0.548/0.506/0.515 | 0.190/0.458/0.469 | 0.650/0.435/0.471 |
| Horizontal Translation | 0.287/0.402/0.459 | 0.140/0.043/0.101 | 0.557/0.504/0.526 | 0.196/0.451/0.475 | 0.670/0.424/0.495 |
| Vertical Shear | 0.397/0.459/0.508 | 0.143/0.079/0.134 | 0.550/0.516/0.547 | 0.223/0.473/0.501 | 0.718/0.483/0.543 |
| Horizontal Shear | 0.398/0.458/0.505 | 0.148/0.084/0.132 | 0.551/0.518/0.546 | 0.223/0.472/0.499 | 0.722/0.487/0.540 |
| Contrast | 0.731/0.496/0.525 | 0.616/0.123/0.146 | 0.787/0.539/0.551 | 0.387/0.507/0.519 | 0.900/0.517/0.545 |
| Invert | 0.644/0.504/0.534 | 0.508/0.142/0.172 | 0.730/0.546/0.563 | 0.322/0.512/0.523 | 0.864/0.533/0.569 |
| Typicality Test | 0.332/0.415/0.460 | 0.004/0.024/0.057 | 0.500/0.500/0.505 | 0.228/0.476/0.498 | 0.595/0.410/0.459 |

| Target: CelebA, OOD: SVHN, CIFAR-10, CIFAR-100 | | | | | |
|---|---|---|---|---|---|
| Method | AUROC | TNR(95% TPR) | Detection accuracy | AUPR in | AUPR out |
| SVD-RND (proposed) | **0.999**/0.963/0.964 | **0.999**/0.897/0.897 | **0.998**/0.928/0.928 | **0.999**/0.981/0.981 | **0.998**/0.941/0.943 |
| DCT-RND (proposed) | 0.997/0.854/0.879 | 0.989/0.491/0.587 | 0.989/0.771/0.797 | 0.996/0.936/0.945 | 0.997/0.736/0.794 |
| GB-RND (proposed) | 0.997/0.824/0.845 | 0.994/0.455/0.526 | 0.994/0.748/0.762 | 0.996/0.918/0.926 | **0.998**/0.694/0.750 |
| RND | 0.410/0.743/0.741 | 0.067/0.231/0.253 | 0.512/0.681/0.678 | 0.439/0.883/0.879 | 0.459/0.500/0.523 |
| GPND | 0.407/0.742/0.737 | 0.084/0.230/0.250 | 0.536/0.680/0.680 | 0.461/0.879/0.870 | 0.478/0.502/0.520 |
| Flip | 0.402/0.898/0.906 | 0.055/0.728/0.750 | 0.507/0.840/0.851 | 0.440/0.946/0.948 | 0.447/0.830/0.845 |
| Rotate | 0.974/**0.979/0.982** | 0.950/**0.937/0.945** | 0.964/**0.952/0.956** | 0.950/**0.989/0.991** | 0.981/**0.964/0.969** |
| Vertical Translation | 0.964/0.961/0.964 | 0.930/0.887/0.897 | 0.952/0.923/0.926 | 0.934/0.979/0.980 | 0.975/0.941/0.946 |
| Horizontal Translation | 0.955/0.940/0.949 | 0.894/0.874/0.889 | 0.929/0.920/0.926 | 0.926/0.963/0.968 | 0.967/0.922/0.932 |
| Vertical Shear | 0.974/0.960/0.964 | 0.940/0.883/0.897 | 0.954/0.921/0.929 | 0.962/0.978/0.981 | 0.977/0.940/0.947 |
| Horizontal Shear | 0.951/0.963/0.964 | 0.897/0.885/0.888 | 0.932/0.920/0.922 | 0.922/0.982/0.982 | 0.963/0.936/0.945 |
| Contrast | 0.848/0.772/0.778 | 0.701/0.295/0.329 | 0.826/0.702/0.707 | 0.766/0.897/0.898 | 0.885/0.550/0.568 |
| Invert | 0.860/0.908/0.903 | 0.766/0.740/0.721 | 0.864/0.847/0.839 | 0.752/0.949/0.947 | 0.910/0.871/0.861 |
| Typicality Test | 0.484/0.749/0.749 | 0.064/0.245/0.274 | 0.516/0.691/0.690 | 0.500/0.878/0.874 | 0.491/0.518/0.545 |

## B  RANDOM NETWORK DISTILLATION

We use RND (Burda et al., 2019) as the base model of our OOD detector. RND consists of a trainable predictor network $f$, and a randomly initialized target network $g$. The predictor network is trained to minimize the $l_2$ distance against the target network on training data. The target network $g$ remains fixed throughout the training phase.

$$f^* = \arg\min_{f} \Sigma_{x \in D_{\text{train}}} \|f(x) - g(x)\|_2^2 \tag{5}$$

Then, for unseen test data $x$, RND outputs $\|f(x) - g(x)\|_2^2$ as an uncertainty measure. The main intuition of the RND is to reduce the distance between $f$ and $g$ only on the in-distribution, hence naturally distinguish between the in-distribution and the OOD.

We employ RND for our base OOD detector due to its simplicity over generative models. Also, RND has already shown to be effective in novelty detection on MNIST dataset (Burda et al., 2019).

In RND (Burda et al., 2019), the predictor network $f$ has two more layers than the target network $g$, where $g$ consists of 3 convolution layers and a fully connected layer. In our experiments, we set $g$ as the first 33 layers of ResNet34 without ReLU activation in the end. $f$ is constructed by appending two sequential residual blocks. The output size of each residual block is 1024 and 512. We also discard ReLU activation in the second residual block to match the output layer of $g$.

## C  DATA PREPROCESSING, NETWORK SETTINGS, PARAMETER SETTINGS FOR MAIN EXPERIMENT

To make the OOD detection task harder, we reduce each of the training dataset of CelebA, Tiny-ImageNet, and LSUN to 50000 examples and test dataset of CelebA to 26032 examples. For TinyImageNet data, we discard half of the images in each class, resulting in 250 training examples for each of the 200 classes. Reduction in LSUN dataset results in 5000 examples for each of the 10 classes. Also, the first 1000 images of the test OOD data are used for validation. For SVD-RND and all other RND-based detectors, we use the same structure for $f$ and $g$ defined in Appendix B. The number of parameter updates is fixed across the experiments. The Adam optimizer, with a learning rate of $10^{-4}$, is used for RND-based OOD detection methods. The learning rate is annealed from $10^{-4}$ to $10^{-5}$ in half of the training process. For our main experiment, we average the result across two fixed random seeds.

In SVD-RND, DCT-RND, and GB-RND, we used $b_{train} = 1$ for CIFAR-10 and CelebA dataset, and $b_{train} = 2$ for TinyImageNet and LSUN dataset. For SVD-RND, We optimize across $K_1 \in \{18, 20, 22, 24, 25, 26, 27, 28\}$ in the CIFAR-10 and CelebA datasets. For TinyImageNet and LSUN datasets, we optimize over $K_1 \in \{8, 10, 12, 14\}$ and $K_2 \in \{22, 24, 26, 28\}$. In DCT-RND, we define $K_i$ as the number of unpruned signals in the frequency domain. For CIFAR-10 and CelebA datasets, we optimize $K_1$ across $\{4, 8, 12, 14, 16, 20, 24, 28\}$. For TinyImageNet and LSUN datasets, we optimize over $K_1 \in \{20, 24, 28, 32\}$ and $K_2 \in \{40, 44, 48, 52\}$. For Gaussian blurring, we optimize over the shape $(x_i, y_i)$ of the Gaussian kernel. We optimized the parameter over $x_i \in \{1, 3, 5\}$, $y_i \in \{1, 3, 5\}$ for each blurred data. To fix the number of updates, we train SVD-RND, DCT-RND, and GB-RND for 50 epochs in the CIFAR-10 and CelebA datasets, and for 34 epochs for the rest.

For GPND, the settings for the original paper are followed. Furthermore, we optimize the reconstruction loss $\lambda_1$ and adversarial loss $\lambda_2$ for discriminator $D_z$ across $\lambda_1 \in \{8, 9, 10, 11, 12\}$ and $\lambda_2 \in \{1, 2, 3\}$. We choose the parameters with the best validation performance at 100 epochs,

For RND, we trained over 100 epochs.

For geometric transforms, we optimize the magnitude of the shift of shear, horizontal translation and vertical translation methods. We optimize the magnitude of translation across $\{4, 8, 12, 16\}$ and choose the parameter with the best validation performance. Detector is trained for $\lceil \frac{100}{|T|+1} \rceil$ epochs, where $|T|$ is the number of transformations. The number of transformations is 1 in flipping and invert, 2 for horizontal translation, vertical translation, and shear, and 3 for rotation and contrast.

Table 7: Test uncertainty of RND on OOD CIFAR-10 data generated by adding orthogonal noise to the CIFAR-10 data

| Data | Original | Blurred | $\alpha = 5$ | $\alpha = 10$ | $\alpha = 15$ | $\alpha = 20$ |
|---|---|---|---|---|---|---|
| Average Uncertainty($\times 10^{-5}$) | 5.631 | 5.190 | 5.648 | 5.795 | 6.051 | 6.437 |

Finally, for typicality test, we estimated the average test loss of the RND for 50000 training examples. For each test sample, we use the distance between the test loss of the sample and the estimated average loss as the OOD metric.

## D    GENERATING OOD BY ADDING ORTHOGONAL VECTORS

We present the performance of RND on OODs generated by adding vectors orthogonal to the data. To genetrate such OODs, we sample a Gaussian vector $z$ and compute the component of the random vector $z_{orth,x}$ that is orthogonal to the data x.

$$z_{orth,x} = z - \frac{z^T x}{x^T x} x \qquad (6)$$

We scaled the $l_2$ norm of the orthogonal vector $z_{orth,x}$ on each data to be $\alpha\%$ of the $l_2$ norm of the signal. We plot the average uncertainty of RND on the original data, blurred data, and the perturbed data in Table 7. From the 20 independent runs on the perturbed data, we report the case with smallest test uncertainty in Table 7. We varied $\alpha$ from 5 to 20. While blurring reduces the average test uncertainty of RND, adding orthogonal vector to the data incerases the test uncertainty of RND.

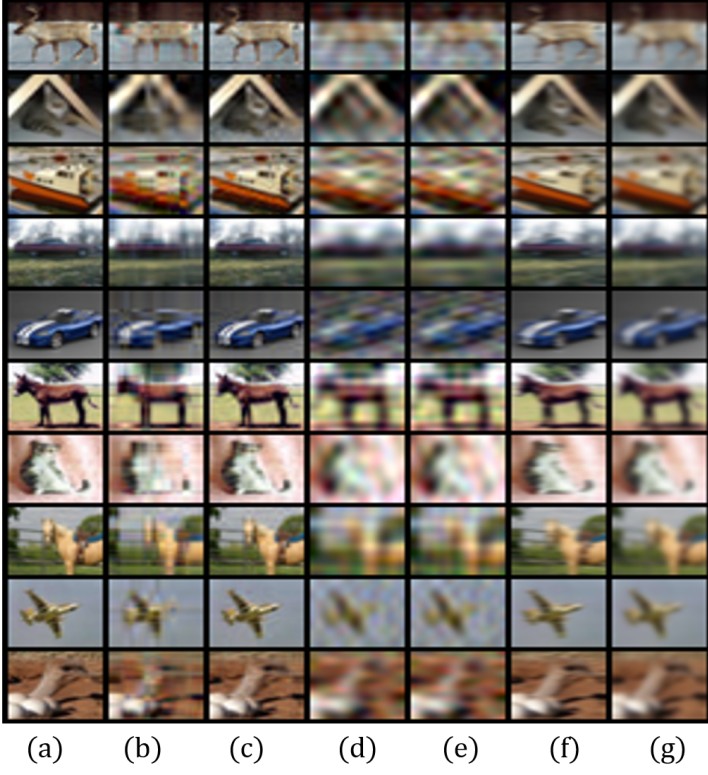

(a)      (b)      (c)      (d)      (e)      (f)      (g)

Figure 5: Sample visualization on the best performing parameters of SVD-RND, DCT-RND, and GB-RND. (a): original CIFAR-10 sample. (b), (c): sample after SVD-RND. (d), (e): sample after DCT-RND. (f), (g): sample after GB-RND.

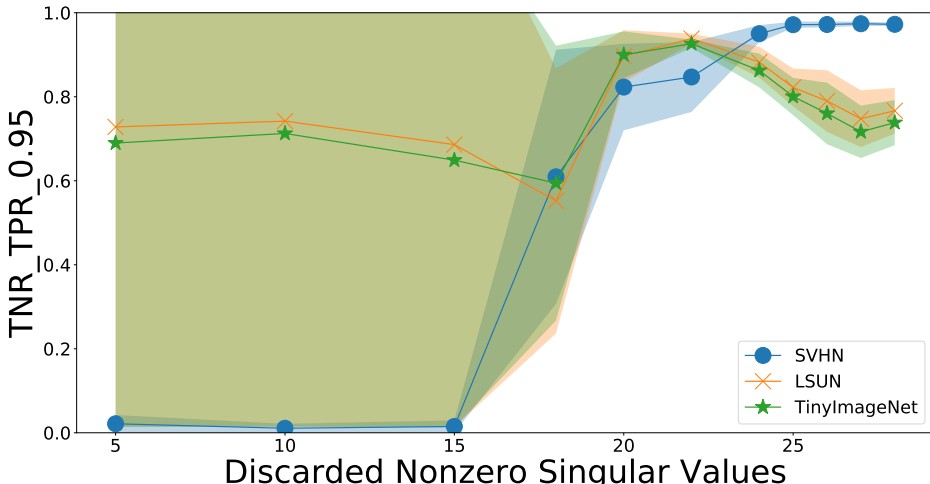

Figure 6: Extended version of Figure 3 (left). When we discard small nonzero eigenvalues for the blurred images, SVD-RND fails to discriminate between original and blurred images.

## E    VISUALIZATION OF DIFFERENT BLURRING TECHNIQUES

For visualization, we plot the CIFAR-10 images and their blurred versions processed by SVD-RND, DCT-RND, and GB-RND in Figure 5. Images in the same column are processed with the same technique. Furthermore, columns (b), (d), (e) are constructed by the best performing parameters of SVD-RND, DCT-RND, and GB-RND on SVHN OOD data. Likewise, (c), (e), (f) are constructed by the best performing parameters of SVD-RND, DCT-RND, and GB-RND on TinyImageNet OOD data.

## F    EXTENDED VERSION OF FIGURE 3

We further extend Figure 3 to analyze the behavior of SVD-RND when small number of singular values are discarded. Therefore, we experiment SVD-RND where $K_1 = 5, 10, 15$ and plot the result in Figure 6.

