# OpenReview forum: "Novelty Detection Via Blurring"
_ICLR.cc/2020/Conference — Accept (Poster)_

### Official Review · AnonReviewer3 · 2019-10-21
**Official Blind Review #3**

**Rating:** 6

**Review:**

UPDATE:
I acknowledge that I‘ve read the author responses as well as the other reviews.
I appreciate the clarifications and improvements made to the paper. I‘ve updated my score to 6 Weak Accept.

####################

This paper presents the idea to use blurred images as regularizing examples to improve out-of-distribution (OOD) detection performance based on Random Network Distillation (RND). The paper proposes to generate sets of such blurred images via Singular Value Decomposition (SVD) on the training images by pruning the lowest K non-zero singular values. The proposed method, SVD-RND, then extends the standard RND objective, which is to train a predictor network f to minimize the L2 loss to the output of some randomly initialized network over the (original) training data, with an additional regularization term that minimizes the L2 loss to the outputs of further multiple randomly initialized networks over the sets of blurred images. In OOD experiments on CIFAR-10, SVHN, TinyImageNet, LSUN, and CelebA, the proposed SVD-RND consistently outperforms baselines and recent competitors which are demonstrated to be vulnerable to blurred images.

I find it hard to make a definitive evaluation for this work at this point and would like to take the authors' responses into account for my final recommendation. The main idea of the paper to generate adversarial examples for training via blurring is rather simple and thus the novelty of this work is somewhat minor. I think the quality of the paper also suffers from many statements in the text that draw too general and too bold conclusions at this point in my mind. The presentation overall is rather unpolished (see comments below). However, I find the empirical results itself quite strong and convincing and think they would make a relevant and significant contribution to the community. I do have questions left open though that need clarification:

(i) I don’t see why different techniques for blurring (SVD, DCT, GB) should lead to such different results as the approach remains conceptually similar. Do you have a reason/intuition why SVD gives the best results? Might SVD just be the easiest to tune method?

(ii) What do you think is the key reason that SVD-RND also appears to generalize too non-blurry OOD samples? Could you elaborate more on the two reasons you give in the paper? (1. RND performance on samples orthogonal to the data; 2. Discrimination between data and its low-rank projection)

(iii) The generation and tuning of multiple sets of blurred images (how many samples per set?) may get quite extensive for large datasets. Could you be specific on the computational cost?

(iv) Might the deep generative models (e.g. GPND) fail to detect blurred images due to insufficient model capacity of the decoder which results in blurry reconstructions? Have you varied the network capacity or latent space dimensionality of such models?

(v) What is the idea behind choosing the log effective rank in such an equidistant manner as proposed?


####################
*Additional Feedback*

*Positive Highlights*
1. SVD-RND shows strong OOD detection performance in OOD experiments on a variety of image dataset combinations (CIFAR-10, SVHN, TinyImageNet, LSUN, and CelebA).
2. The related work includes major works from all the related lines of research (Deep anomaly detection; OOD detection using side information; Adversarial examples/training)
3. Useful hyperparameter selection criterion based on effective rank if no OOD validation data is available.

*Ideas for Improvement*
4. I think the tone of the paper would greatly benefit from not drawing too general conclusions and too bold implications. Keep statements precise and evidence-based. Declare hypotheses as such.
5. I would appreciate a plot showing samples before and after blurring in the appendix to see the most effective degree of blurring. Maybe also compare the different blurring baselines here to see differences.
6. Report std. devs. with your performance results to infer statistical significance.
7. Compare to the specific geometric transforms method as proposed in the paper [3] and not only using those transformations within your RND approach.
8. Add missing deep anomaly detection related work [6, 2, 5, 1].
9. Expand the sensitivity analysis in Figure 3 (left) over a greater range of K. Especially, I would like to also see K = 0 (unblurred, original images) as a sanity check which may only improve over RND due to ensembling over multiple randomly initialized networks.
10. Consider the one-vs-rest anomaly detection evaluation benchmarks from Ruff et al. [4] or Golan and El-Yaniv [3] to further infer the generalization performance of the proposed method.
11. I think the paper spends too much time on introducing previous work. Section 2 Related Work and Section 3 Background might be combined into one section.

*Minor comments*
12. In the abstract: “... VAE or RND are known to assign lower uncertainty to the OOD data than the target distribution.” is a bit strong. Rather “have been observed” etc. This is a working hypothesis in the community, but there is recent work (https://openreview.net/forum?id=Skg7VAEKDS) indicating (at least for VAEs) those are effects of poor model design.
13. In the abstract: “... efficient in test time ...” » “... efficient at test time ...”
14. In the abstract: “... in CelebA domain.” » “... on the CelebA dataset.” or just “... on CelebA.”
15. Section 1: “However, such models show underwhelming performance on detecting OOD, such as detecting SVHN from CIFAR-10. Specifically, generative models assign a higher likelihood to the OOD data than the training data.”. I think those are way too general conclusions at the moment. Rather something like “OOD detection performance of deep generative models has been called into question” and “have been observed to assign ...”.
16. Section 1: “Such results clearly support the degeneracy of deep OOD detection schemes”. Again, I find this way too bold of a statement at this point in time.
17. Section 2: “... a recently proposed paper ...” » “... a recent paper ...”
18. Section 2: “... outlier data independent of OOD data.”. What would outlier data not being out-of- distribution be?
19. Section 2: “Golan et al. (2018) design geometrically transformed data and regularized the classifier ...”. Not regularized. They trained a classifier on labels identified with these transformations.
20. Section 2: “..., resulting in OOD detection in each labeled data” » “..., resulting in OOD detection on labeled data”
21. A subsection title directly following a section title is bad style. A major section should be introduced with a few sentences on what this section is about.
22. Figure 2, right plot: These loss curves are rather strange... Increasing, then sharply decreasing again. Is there a drop in learning rate at epoch 80?
23. Many axis labels are too small and hard to read.
24. In Section 3.3.: “.. in Section 7.2.” » “.. in Section 4.2.”?
25. The definition of the log effective rank in Eq.~2 is weird. It's the entropy over the singular value distribution, i.e.~$H\left(\sigma_1 / \sum_j \sigma_j, \ldots, \sigma_N / \sum_j \sigma_j \right)$. Also, the parameters/notation involved are not introduced.
26. Make clear you apply SVD on single images. The paper alters formulations between data matrix and images...
27. Several instances where citet is used instead of citep.
28. In Table 1: Separate the target column from the three OOD columns more clearly. (e.g. vertical separator, center OOD, target in bold, etc.)
29. In Section 5.1: “area of the region under the ... curve” » “area under the ... curve”.
30. Figure 3: Better explain the plots. Are the three curves the respective target classes? What is the OOD set?
31. Figure 4: Maybe use subfigures with individual titles/captions.
32. Section 6: “For evidence, we fine-tune the classifier ...” » “For evidence, we fine-tune a classifier ...”
33. Plots and Figures are somewhat scattered and not referenced chronologically.
34. Introduce the effective rank at the point where it is used (Section 6.2). Somewhat unclear why to introduce this in Section 3 already.
35. Visually speaking, I find the RND examples in Figure 4 actually more anomalous than the top-anomalous SVD-RND samples (flashy colors, weird angles, borders, high contrast, ...)


####################
*References*
[1] R. Chalapathy and S. Chawla. Deep learning for anomaly detection: A survey. arXiv preprint arXiv:1901.03407, 2019.
[2] H. Choi, E. Jang, and A. A. Alemi. Waic, but why? generative ensembles for robust anomaly detection. arXiv preprint arXiv:1810.01392, 2018.
[3] I. Golan and R. El-Yaniv. Deep anomaly detection using geometric transformations. In NIPS, 2018.
[4] L. Ruff, R. A. Vandermeulen, N. Görnitz, L. Deecke, S. A. Siddiqui, A. Binder, E. Müller, and M. Kloft. Deep one-class classification. In International Conference on Machine Learning, pages 4393–4402, 2018.
[5] L. Ruff, R. A. Vandermeulen, N. Görnitz, A. Binder, E. Müller, K.-R. Müller, and M. Kloft. Deep semi-supervised anomaly detection. arXiv preprint arXiv:1906.02694, 2019.
[6] T. Schlegl, P. Seeböck, S. M. Waldstein, U. Schmidt-Erfurth, and G. Langs. Unsupervised anomaly detection with generative adversarial networks to guide marker discovery. In Proceedings International Conference on Information Processing in Medical Imaging, pages 146–157. Springer, 2017.


**Experience Assessment:**

I have published in this field for several years.

**Review Assessment: Checking Correctness Of Derivations And Theory:**

I carefully checked the derivations and theory.

**Review Assessment: Checking Correctness Of Experiments:**

I carefully checked the experiments.

**Review Assessment: Thoroughness In Paper Reading:**

I read the paper thoroughly.

---

> ### Author Response · Authors · 2019-11-14
> **Extended Response**
>
> 8. Compare to the specific geometric transforms method as proposed in the paper [3] and not only using those transformations within your RND approach.
>
> -> Paper [3] has also done an evaluation in CIFAR-10 : TinyImageNet domain. In the experiment, they used data label information to train the network. Yet, we tested the scheme by the official implementations of the paper.
>
> The result shows AUROC/AUPR_IN/AUPR_OUT of 0.951/0.958/0.936 in CIFAR-10: TinyImageNet domain, and AUROC/AUPR_IN/AUPR_OUT of 0.934/0.945/0.913 in CIFAR-10: LSUN domain, which underperforms over SVD-RND. We plan to add the additional results to the unlabelled setting in the final version.
>
> 9. Add missing deep anomaly detection related work [1,2,4,5].
>
> -> We added the works in our revised version.
>
> 10. Expand the sensitivity analysis in Figure 3 (left) over a greater range of K. Especially, I would like to also see K = 0 (unblurred, original images) as a sanity check which may only improve over RND due to ensembling over multiple randomly initialized networks.
>
> -> We added the works in Appendix F of our revised version. However, we note that setting K=0 does not improve the results since the objective then becomes to train a network on two data, which have the same inputs and different outputs.
> In other words, the predictor network is trained to match different target networks on the same inputs. We think t
>
> 11. I think the paper spends too much time on introducing previous work. Section 2 Related Work and Section 3 Background might be combined into one section.
>
> -> As suggested, we changed the layout in the revised version.
>
> 12. Figure 2, right plot: These loss curves are rather strange... Increasing, then sharply decreasing again. Is there a drop in learning rate at epoch 80?
>
> -> We note that we did not use any regularization technique in RND. We only dropped the learning rate after 50 epochs.
>
> 13. Many axis labels are too small and hard to read.
>
> -> We changed the size of the axis labels in Figure 1.
>
> 14. Figure 3: Better explain the plots. Are the three curves the respective target classes? What is the OOD set?
>
> -> We appended additional explanations in the caption of Figure 3.
>
> 15. Introduce the effective rank at the point where it is used (Section 6.2). Somewhat unclear why to introduce this in Section 3 already.
>
> -> We addressed this comment in the revised version.
>
> 16.  We further addressed the minor comments in the revised version of the paper.
>
> Again, thanks for the valuable comments that helped to improve this paper.
>
>
> *References*
> [1] R. Chalapathy and S. Chawla. Deep learning for anomaly detection: A survey. arXiv preprint arXiv:1901.03407, 2019.
> [2] H. Choi, E. Jang, and A. A. Alemi. Waic, but why? generative ensembles for robust anomaly detection. arXiv preprint arXiv:1810.01392, 2018.
> [3] I. Golan and R. El-Yaniv. Deep anomaly detection using geometric transformations. In Neural Information Processing Systems, 2018.
> [4] L. Ruff, R. A. Vandermeulen, N. Görnitz, A. Binder, E. Müller, K.-R. Müller, and M. Kloft. Deep semi-supervised anomaly detection. arXiv preprint arXiv:1906.02694, 2019.
> [5] T. Schlegl, P. Seeböck, S. M. Waldstein, U. Schmidt-Erfurth, and G. Langs. Unsupervised anomaly detection with generative adversarial networks to guide marker discovery. In Proceedings International Conference on Information Processing in Medical Imaging, pages 146–157. Springer, 2017.

---

> ### Author Response · Authors · 2019-11-14
> **Response to Reviewer 3**
>
> We thank the reviewer for extensive and worthy feedback.
>
> 1. I don’t see why different techniques for blurring (SVD, DCT, GB) should lead to such different results as the approach remains conceptually similar. Do you have a reason/intuition why SVD gives the best results? Might SVD just be the easiest to tune method?
>
> -> We first note that the performance of Table 2. was conducted on the small size of the hyperparameter search, which cannot cover the whole parameter space of DCT-RND and GB-RND. Potentially, when we increase b_train or hyperparameter search space, GB-RND or DCT-RND can outperform SVD-RND. For example, GB-RND shows similar results to SVD-RND on LSUN : SVHN, or TinyImageNet : SVHN domain.  Therefore, although the result in Table 2 favors SVD-RND, we hesitate to assert that there is something special about SVD-RND.
> Rather, we just think SVD-RND is much easier to optimize than DCT-RND or GB-RND, and this contributed to the success of SVD-RND in Table 2.
>
> 2. What do you think is the key reason that SVD-RND also appears to generalize too non-blurry OOD samples? Could you elaborate more on the two reasons you give in the paper? (1. RND performance on samples orthogonal to the data; 2. Discrimination between data and its low-rank projection)
>
> -> Reviewer 1 also mentioned this aspect and we conducted an ablation study on reasoning #2. We conducted the ablation by setting two cases: adversarial images that have the same rank to the blurred images, and adversarial images that have the same effective rank to the blurred images. The result of those ablations was much worse than SVD-RND. Therefore, we hypothesize that blurred images function as "projection" helps SVD-RND to achieve better OOD detection.
>
> 3. The generation and tuning of multiple sets of blurred images (how many samples per set?) may get quite extensive for large datasets. Could you be specific on the computational cost?
>
> -> We generate each blurred image as equal to the size of the training data, which is 50000 on the experiment. To be fair, we scaled the training epochs proportional to $\frac{1}{b_{train}+1}$.
>
> In the training phase, we observed that generating blurred images compared to geometric transforms does not contribute much to the computation cost. Rather, based on our setting, we employ an additional target network per blurred image and found that evaluation of the target network is the major factor in the computation cost when $b_{train}$ increases.
>
> Specifically, when $b_{train}=1$, 0.255 seconds are taken per batch. When $b_{train}=2$, 0.294 seconds are taken per batch. When $b_{train}=3$, 0.333 seconds are taken per batch on average. Therefore, we can extrapolate that the load of 0.039 seconds is appended for computation cost when b_train increases by 1.
>
> 4. Might the deep generative models (e.g. GPND) fail to detect blurred images due to insufficient model capacity of the decoder which results in blurry reconstructions? Have you varied the network capacity or latent space dimensionality of such models?
>
> -> We have tried some ablation studies on GPND in CIFAR-10 : (SVHN, LSUN, TinyImageNet ) domain.
>
> 1) First, we doubled the size of the latent dimension, and this gave worse results.
> 2) We changed the structure of encoder to ResNet34 and this gave TNR(at 95% TPR) of 0.052/0.807/0.692.
> 3) We changed the capacity of the network by doubling the output size of each layer of the encoder, decoder, and discriminator. This gave TNR of 0.053/0.820/0.725.
>
> In conclusion, through our ablation analysis, performance on SVHN : (LSUN, TinyImageNet) improved by varying the network capacity of GPND. However, we failed to improve OOD detection performance in CIFAR-10 : SVHN domain.
>
> 5. What is the idea behind choosing the log effective rank in such an equidistant manner as proposed?
>
> -> By setting target log effective rank equidistantly, we wanted to minimize the worst-case log effective rank difference between the potential OOD and the blurred data chosen by our heuristic.
>
> 6. I think the tone of the paper would greatly benefit from not drawing too general conclusions and too bold implications. Keep statements precise and evidence-based. Declare hypotheses as such.
>
> -> Thanks for the feedback. We have modified strong assertions in the revised version.
>
> 7. I would appreciate a plot showing samples before and after blurring in the appendix to see the most effective degree of blurring. Maybe also compare the different blurring baselines here to see differences.
>
> -> We also appended the CIFAR-10 sample before and after blurring in Appendix E. We plotted with the best performing parameters of SVD-RND, DCT-RND, and GB-RND.

---

### Official Review · AnonReviewer2 · 2019-10-23
**Official Blind Review #2**

**Rating:** 6

**Review:**

Summary: They tackle the problem of out-of-data distribution by leveraging RND applied to data augmentations. They train a model f(x) to match the outputs of g_i(aug_i(x)), where g_i is a random network and aug_i is a particular type of augmentation. An example with high error in this task is treated as an out-of-distribution example. This work focuses on exploring blurring through SVD, where the smallest K singular values are set to 0, and K varies between different aug_i calls. They find that their method of consistently can achieve strong detection rates across multiple target-dataset pairs.

Comments:
* The experimental results in this work are impressive, which introduces many more questions.
* The model used for f and g is not mentioned in the text.
* Figure 4 (left) suggests that the SVD-RND performs about the same between 10K and 50K examples. The level of robustness is surprising, but doesn’t seem to square with intuition that more data ought to help. How little data can be used? In other words, extend the graph to the left.
* The geometric transforms baseline is not fair, since SVD-RND uses multiple SVD transforms (b_train > 1) whereas the geometric transforms only have one. Please run a model with all the geometric transforms. This result is important for understanding whether the gains come from the particular transform (SVD) or the number of transforms used.
* Following the spirit of the previous comment, what other data augmentations can be used in place of SVD? Typical image classification pipelines use a large variety of augmentations. I would suggest taking some augmentations from AutoAugment [1] and running RND on top of them.
* An experiment that is missing is RND trained on blurred images. Is the blurring itself the important component, or is having multiple different heads important?
* In general, I am confused about how a single head RND does not converge to 0 loss by learning the weights of g. This seems to be a simple optimization problem. The original RND paper avoided this problem by also using the network to learn a policy, but this does not exist in this approach.
* Furthermore, a comparison with Ren et al. [2] and Nalisnick et al. [3] would be useful. [2] also uses data augmentation to create a background model that is compared against the real model. One can probably simulate this approach by comparing the error rates of each head of RND.

In general, this work seems promising, but lacks proper ablations that elucidate what components of the method are important. I am happy to increase my score if the experiments suggests are added to the work.

[1] AutoAugment: Learning Augmentation Policies from Data. Ekin D. Cubuk, Barret Zoph, Dandelion Mane, Vijay Vasudevan, Quoc V. Le
[2] Likelihood Ratios for Out-of-Distribution Detection. Jie Ren, Peter J. Liu, Emily Fertig, Jasper Snoek, Ryan Poplin, Mark A. DePristo, Joshua V. Dillon, Balaji Lakshminarayanan
[3] Detecting Out-of-Distribution Inputs to Deep Generative Models Using Typicality. Eric Nalisnick, Akihiro Matsukawa, Yee Whye Teh, Balaji Lakshminarayanan

**Experience Assessment:**

I have read many papers in this area.

**Review Assessment: Checking Correctness Of Derivations And Theory:**

N/A

**Review Assessment: Checking Correctness Of Experiments:**

I carefully checked the experiments.

**Review Assessment: Thoroughness In Paper Reading:**

I read the paper thoroughly.

---

> ### Author Response · Authors · 2019-11-14
> **Extended Response**
>
> 7. Furthermore, a comparison with Ren et al. [2] and Nalisnick et al. [3] would be useful. [2] also uses data augmentation to create a background model that is compared to the real model.
> One can probably simulate this approach by comparing the error rates of each head of RND.
>
> -> We thank the reviewer for introducing the two papers. We experimented with the typicality test introduced in [3] and added the results in the revised version.
>
> We also experimented with [2] by training the background RND model on the mutated data, and setting the test metric as "test loss of original network - test loss of background model". However, the result was worse than RND with TNR( at 95% TPR) of 0.009/0.350/0.368 in CIFAR-10 : (SVHN, LSUN, TinyImageNet) domain and 0.040/0.125/0.105 at CelebA : (SVHN, CIFAR-10, CIFAR-100) domain. One possible explanation for this underwhelming behavior of the background model is that RND is not an autoregressive model like PixelCNN and therefore less suitable to learn pixel mutation.
>
> Again, we appreciate the valuable comments that improved the paper.
>
> [1] AutoAugment: Learning Augment Policies from Data. Ekin D. Cubuk, Barret Zoph, Dandelion Mane, Vijay Vasudevan, and Quoc V. Le. In IEEE Conference on Computer Vision and Pattern Recognition, 2019.
> [2] Likelihood Ratios for Out-of-Distribution Detection. Jie Ren. Peter J. Liu, Emily Fertig, Jasper Snoek, Ryan Poplin, Mark A. DePristo, Joshua V. Dillon, and Balaji Lakshminarayanan. arXiv Preprint: arXiv:1906:02845, 2019.
> [3] Detecting Out-of-Distribution Inputs to Deep Generative Models Using Typicality. Eric Nalisnick, Akihiro Matsukawa, Yee Whye Teh, and Balaji Lakshminarayan. arXiv Preprint: arXiv:1906:02994, 2019.

---

> ### Author Response · Authors · 2019-11-14
> **We incorporated some baselines for the experiment**
>
> We first thank the reviewer for the feedback that helped to improve the paper.
>
> 1. The model used for f and g is not mentioned in the text.
>
> -> We modified the style of the paper and referenced the structure in Appendix B.
>
> 2. Figure 4 (left) suggests that the SVD-RND performs about the same between 10K and 50K examples. The level of robustness is surprising, but doesn’t seem to square with intuition that more data ought to help. How little data can be used? In other words, extend the graph to the left.
>
> -> We extended the graph where 2000,4000,6000,8000 training data are available. When 8000 training data is available, the performance of SVD-RND drops but still outperforms the performance of RND with 50000 training data. When the data size is smaller than 8000, the performance drop is huge. We modified Figure 4 to present the phenomenon mentioned above.
>
> 3.  The geometric transforms baseline is not fair, since SVD-RND uses multiple SVD transforms ($b_{train}$ > 1) whereas the geometric transforms only have one. Please run a model with all the geometric transforms. This result is important for understanding whether the gains come from the particular transform (SVD) or the number of transforms used.
>
> -> First, we have combined the model with geometric transformations in the RND framework. We experimented by merging rotation, flip, and translation, and this results in b_train=4*3*2-1=23. Such a combination showed TNR(at 95% TPR) of 0.181/0.182/0.199 in CIFAR-10 : (SVHN, LSUN, TinyImageNet) domain. This shows that combining different transforms to one does not always improve OOD detection performance.
>
> Also, we want to clarify that rotation or translation employs multiple transforms for training. For example, in rotation, we assign different target network to each data rotated by 90, 180, and 270 degrees. Therefore, rotation already has $b_{train}=3$.
>
> 4.  Following the spirit of the previous comment, what other data augmentations can be used in place of SVD? Typical image classification pipelines use a large variety of augmentations. I would suggest taking some augmentations from AutoAugment and running RND on top of them.
>
> -> We further experimented with the contrast, shear, and invert introduced in [1], and appended the results on the revised version.
>
> 5.  An experiment that is missing is RND trained on blurred images. Is the blurring itself the important component, or is having multiple different heads important?
>
> -> We also experimented RND trained on blurred images in the CIFAR-10 : (SVHN, LSUN, TinyImageNet) domain with varying K. This strategy showed the TNR(at 95% TPR) of 0.021/0.809/0.767, which showed slightly improved result compared to RND but still underperforms over SVD-RND. Therefore, we hypothesize that "teaching the network to discriminate between blurred images and original images helped OOD detection".
>
> 6. In general, I am confused about how a single head RND does not converge to 0 loss by learning the weights of g. This seems to be a simple optimization problem. The original RND paper avoided this problem by also using the network to learn a policy, but this does not exist in this approach.
>
> -> First, we set that each target network in the experiment also corresponds to the complex function. Therefore, we expect that much strict learning rate scheduling and more training epoch will be required to better convergence.
>
> Also, we reviewed the original RND paper and found that they do not update the predictor network via the policy loss. The paper updates the predictor network only on the $l_{2}$ loss, which is the same as us. We refer to the 14th page of  https://openreview.net/pdf?id=H1lJJnR5Ym for the reference.

---

### Official Review · AnonReviewer1 · 2019-10-24
**Official Blind Review #1**

**Rating:** 6

**Review:**

This paper proposed a method called SVD-RND to solve the out-of-distribution (OOD) detection problem. The proposed SVD-RND is under the Random Network Distillation (RND) framework, but make use of blurred images as adversarial samples. SVD-RND outperforms state-of-the-art methods in several OOD tasks.

In general, this paper is well-structured. It is not difficult to understand the problem this paper focuses on and the proposed method. I believe the proposed method is interesting, and I have not seen a similar approach before. It is a simple method, but it achieves excellent performance in multiple OOD tasks.

Although the authors try to explain why SVD-RND performs well, I am not entirely convinced and believe that more investigations might be necessary. Are the images from different datasets of similar average effective rank? I am also wondering how the proposed model will perform if we use other images as the adversarial examples. For example, we can use random low-rank images, which can be generated via Equation (3) by first randomly sampling $\sigma_{jt}$, $u_{jt}$, and $v_{jt}$, and then let a certain number of singular values be zero.

In summary, I am inclined to accept this paper because it proposes a simple method that gives high performance. However, I would suggest the authors include more ablation studies to help the readers understand why the proposed method works.

Minor:
I suggest the authors explicitly state that we do not update the network $g$ in Equation (1) and (4), such that the readers are less likely to be confused. The authors might also need to briefly explain why RND works.

Is a specific strategy applied to initialize the network $g$ randomly? Are weights in $g$ initialized using Gaussian distribution, uniform distribution, or via other initialization strategies?

In Table 1, all the datasets in columns 2, 3, and 4 are OOD samples, correct?

To ensure that the TPR  is $95\%$ as described in Table 2, we need to tune the threshold for $|| f(x) - g_0(x) ||_2^2$, right? If a sample $x$ gives a $|| f(x) - g_0(x) ||_2^2$ that is higher than the threshold, it is considered as an OOD sample, correct?


**Experience Assessment:**

I do not know much about this area.

**Review Assessment: Checking Correctness Of Derivations And Theory:**

I carefully checked the derivations and theory.

**Review Assessment: Checking Correctness Of Experiments:**

I assessed the sensibility of the experiments.

**Review Assessment: Thoroughness In Paper Reading:**

I read the paper thoroughly.

---

> ### Author Response · Authors · 2019-11-14
> **Response to Reviewer1**
>
> We first thank you for the extensive feedback that helped to improve our paper.
>
> 1. Are the Images from different datasets of similar average effective rank?
>
> -> No. The different dataset has a different average effective rank in general. Furthermore, effective rank has some correlation to the pathologies observed in [1]. For example, SVHN has an average effective rank of 5.13, which is smaller than the average effective rank of CIFAR-10, which is 10.57. LSUN : CIFAR10 or ImageNet : CIFAR10 also possess a similar relationship.
>
> 2. What happens If we use other images as adversarial examples?
>
> -> We conducted two experiments related to these questions. First, we tested random low-rank images generated by random sampling of $u_{jt}$,$\sigma_{jt}$, and $v_{jt}$ as suggested by the reviewer. Setting such random low-rank images as adversaries showed TNR( at 95% TPR) of 0.015/0.691/0.654 in CIFAR-10 : (SVHN, LSUN, TinyImageNet) which is worse than the result on RND.
>
> Furthermore, we experimented by setting the random images that have the same effective rank as the blurred images. We did this by preserving $\sigma_{jt}$ as same as the blurred images and only sampling $u_{jt}$ and $v_{jt}$. Setting such random low effective rank images showed TNR(at 95% TPR) of 0.025/0.755/0.715 in CIFAR-10 : (SVHN, LSUN, TinyImageNet) domain.
>
> Therefore, from such ablation analysis, we can infer that SVD-RND improves over RND since blurred images act as low-rank "projection".
>
> 3. I suggest the authors explicitly state that we do not update the network g in equation (1) and (4), such that the readers are less likely to be confused.
>
> -> We further stated in the main algorithm and appendix B in the revised version that we do not update the target network during the training phase.
>
> 4. The authors might also need to briefly explain why RND works
>
> -> We explained in appendix B in the revised version that the main intuition of RND is to reduce the distance between f and g only on the target distribution, therefore naturally threshold between target and OOD distribution.
>
> 5. Is a specific applied to initialize the network g randomly? Are weights in g initialized using Gaussian distribution, uniform distribution, or via other initialization strategies?
>
> -> We did not perform any explicit initialization strategies for network g and f.  We used the default initialization.
>
> 6. In Table1, all the datasets in columns 2,3,4 are OOD samples, correct?
>
> -> Correct. We modified Table 1 for better visualization.
>
> 7. To ensure that the TPR is as described in Table 2, ...., If a sample gives a  that is higher than the threshold, it is considered as an OOD sample, correct?
>
> -> Yes.
>
> Again, we appreciate the valuable comments that improved the paper.
>
> [1] Do Deep Generative Models Know What They Don't Know? Eric Nalisnick, Akihiro Matsukawa, Yee Whye Teh, Dilan Gorur, and Balaji Lakshminarayanan. In International Conference on Learning Representations, 2019.

---

### Author Response · Authors · 2019-11-14
**Upload of the Revised Version**

First, we thank all the reviewers for the helpful comments. We uploaded the revised version.
We summarize some major changes.

1. We added some baselines for our main experiment. First, we added some augmentation strategies (invert, contrast, and shear) in [1]. Also, we added the typicality test in [2] for comparison.
2. We extended Figure 4 for a smaller number of data samples. Also, we extended Figure 3 for a smaller number of $K_{1}$ in Appendix F.
3. We added a plot showing samples before and after blurring in Appendix E.
4. We deleted the background section and merged the problem formulation with related works.

Again, we thank the reviewers for their extensive feedback.

[1] AutoAugment: Learning Augment Policies from Data. Ekin D. Cubuk, Barret Zoph, Dandelion Mane, Vijay Vasudevan, and Quoc V. Le. In IEEE Conference on Computer Vision and Pattern Recognition, 2019.
[2] Detecting Out-of-Distribution Inputs to Deep Generative Models Using Typicality. Eric Nalisnick, Akihiro Matsukawa, Yee Whye Teh, and Balaji Lakshminarayan. arXiv Preprint: arXiv:1906:02994, 2019.

---

### Decision · Program_Chairs · 2019-12-19

**Decision:**

Accept (Poster)

**Comment:**

The paper proposes a new method for out-of-distribution detection by combining random network distillation (RND) and blurring (via SVD). The proposed idea is very simple but achieves strong empirical performance, outperforming baseline methods in several OOD detection benchmarks. There were many detailed questions raised by the reviewers but they got mostly resolved, and all reviewers recommend acceptance, and this AC agrees that it is an interesting and effective method worth presenting at ICLR.